# SELFOOD: Self-Supervised Out-Of-Distribution Detection via Learning to Rank

**Dheeraj Mekala**[♠,◇]     **Adithya Samavedhi**[♠,◇]     **Chengyu Dong**[◇]     **Jingbo Shang**[◇,♡,*]

[◇]University of California San Diego

[♡] Halıcıoğlu Data Science Institute, University of California San Diego

{dmekala, asamavedhi, cdong, jshang}@ucsd.edu

## Abstract

Deep neural classifiers trained with cross-entropy (CE) loss often suffer from poor calibration, necessitating the task of out-of-distribution (OOD) detection. Traditional supervised OOD detection methods require expensive manual annotation of in-distribution and OOD samples. To address the annotation bottleneck, we introduce SELFOOD, a self-supervised OOD detection method that requires only in-distribution samples as supervision. We cast OOD detection as an inter-document intra-label (IDIL) ranking problem and train the classifier with our pairwise ranking loss, referred to as IDIL loss. Specifically, given a set of in-distribution documents and their labels, for each label, we train the classifier to rank the softmax scores of documents belonging to that label to be higher than the scores of documents that belong to other labels. Unlike CE loss, our IDIL loss reaches zero when the desired confidence ranking is achieved and gradients are backpropagated to decrease probabilities associated with incorrect labels rather than continuously increasing the probability of the correct label. Extensive experiments with several classifiers on multiple classification datasets demonstrate the effectiveness of our method in both coarse- and fine-grained settings.

## 1 Introduction

Deep neural networks (DNNs) are ubiquitously used for text classification (Liu et al., 2019; Devlin et al., 2019; Yang et al., 2019; Brown et al., 2020). However, they are generally poorly calibrated, resulting in erroneously high-confidence scores for both in-distribution and out-of-distribution (OOD) samples (Szegedy et al., 2013; Nguyen et al., 2015; Guo et al., 2017; Mekala et al., 2022a). Such poor calibration makes DNNs unreliable, and OOD detection task vital for the safe deployment of

---

♠ Equal Contribution
∗ Jingbo Shang is the corresponding author.

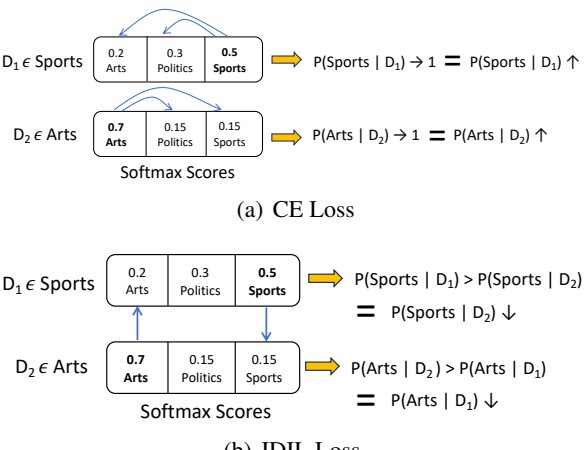

(a) CE Loss

(b) IDIL Loss

Figure 1: CE Loss and IDIL loss optimization for documents $D_1 \in$ *Sports* and $D_2 \in$ *Arts*. CE loss *increases* scores corresponding to the *Sports* for $D_1$ and *Arts* for $D_2$, implying an intra-document comparison. Instead, IDIL loss compares softmax scores in an inter-document intra-label fashion where it *reduces* scores corresponding to *Sports* for $D_2$ to be less than that of $D_1$.

deep learning models in safety-critical applications (Moon et al., 2020).

Traditional supervised OOD detection methods (Hendrycks et al., 2018; Larson et al., 2019; Kamath et al., 2020; Zeng et al., 2021b) assume access to high-quality manually annotated in-distribution and OOD samples. However, this requires extensive annotation of OOD samples belonging to diverse distributions, which is expensive to obtain. Moreover, text classifiers are ideally desired to be more confident on in-distribution samples than OOD samples. However, the poor calibration of DNN precludes this phenomenon.

To address these problems, we propose SELFOOD, a self-supervised OOD detection framework that requires only in-distribution samples as supervision. To adhere to the aforementioned ranking constraint, we formulate OOD detection as an inter-document intra-label (IDIL) ranking problem and train the classifier using our pairwise ranking loss, referred to as IDIL loss. As

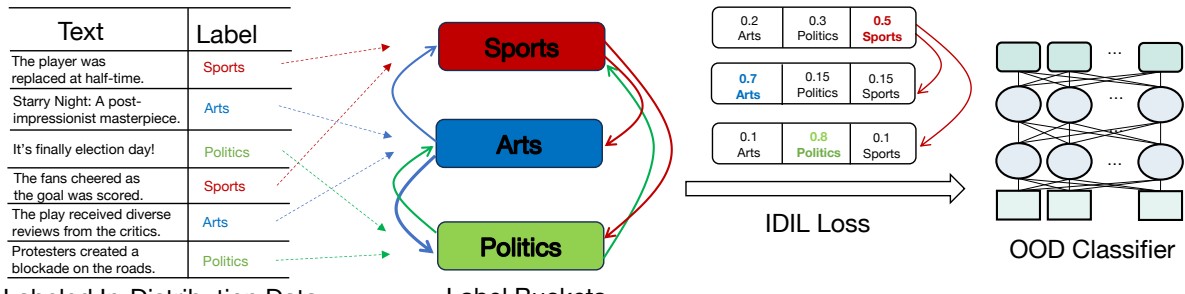

Figure 2: SELFOOD is a self-supervised framework that requires only annotated in-distribution data to train the OOD classifier. Firstly, we bucket documents based on their annotated label (in dotted lines). Then, we compare each document in a bucket with all documents in other buckets to compute IDIL loss (in solid lines). Finally, we backpropagate gradients to decrease scores associated with incorrect labels during the training of the OOD classifier.

shown in Figure 1(a), text classifiers are generally trained using cross-entropy (CE) loss (Good, 1952) in an intra-document fashion where for each document, the classifier is trained to distinguish between different labels by maximizing the score corresponding to the correct label. Instead, in our method, as shown in Figure 1(b), we propose to train in an inter-document, intra-label fashion where for each label, we train the model to rank the considered label probability score in documents belonging to the label higher compared to those not belonging to it. As OOD documents generally do not belong to any in-distribution label, we hypothesize such explicit learning to rank translates to accurately distinguishing OOD from in-distribution.

Moreover, minimizing CE loss involves continuous optimization to increase the probability of the correct label over the other labels, making the classifier overconfident (Wei et al., 2022). Instead, in our method, our IDIL loss function becomes zero once the desired ranking is achieved, and during training, we backpropagate gradients to decrease probabilities associated with incorrect labels rather than increasing the probability of the correct label. Theoretically, the perfect model trained using the CE loss with $0$ training loss is a solution to our ranking problem, however, in Section 5.1, we empirically show that our ranking objective leads to a different solution, demonstrating the importance of the optimization procedure. Finally, it is important to note that our ranking formulation, loss function, and the self-supervised training strategy are specifically designed to improve the performance of OOD detection rather than classification accuracy.

We present our framework in Figure 2. Given a set of in-distribution documents and corresponding labels as input, we bucket documents belonging to each label and train the classifier to rank the probabilities using our IDIL loss function. Specifically, for each document in a label bucket, we pair up with all documents in every other bucket and compute IDIL loss.

Our contributions are summarized as follows:
- We propose SELFOOD, a novel self-supervised method to train an OOD detection model without any OOD samples as supervision.
- We formulate OOD detection as an inter-document intra-label ranking problem and optimize it using our IDIL ranking loss.
- We perform extensive experiments on multiple text classification datasets to demonstrate the effectiveness of our method in OOD detection.
- We release the code on Github[1].

## 2 Related Work

Traditional supervised methods cast the OOD detection as classification with binary labels (Kamath et al., 2020), with one additional label for unseen classes (Fei and Liu, 2016). Since manually annotating documents is expensive (Mekala and Shang, 2020; Mekala et al., 2022b), recent works have deviated from requiring such extensive annotations and leveraging various distance metrics to detect OOD samples. Leys et al. (2018); Xu et al. (2020) use Mahalanobis distance as a post-processing technique to identify OOD samples. These methods use distance-based scoring functions along with the intermediate model layer features to determine an OOD score. Lee et al. (2018); Hsu et al. (2020) use a similar distance metric called ODIN to detect

---

[1] https://github.com/dheeraj7596/SELFOOD

OOD images. An alternate approach to compensate for the lack of OOD training data involves generating pseudo-OOD data for training. Ouyang et al. (2021) propose a framework to generate high-quality OOD utterances and importance weights by selectively replacing phrases in in-domain samples. Zhan et al. (2021) generate pseudo-OOD samples for the task of intent detection using self-supervision. Zhou et al. (2021); Zeng et al. (2021a) introduce self-supervised approaches to OOD detection using a contrastive learning framework. They suggest fine-tuning transformers using a margin-based contrastive loss to learn text representations (Mekala et al., 2021) for OOD classification. Vyas et al. (2018); Li et al. (2021) treat a part of in-domain data as OOD samples as an alternate self-supervised approach. Further, Wu et al. (2022a) use a Reassigned Contrastive Loss (RCL) along with an adaptive class-dependent threshold mechanism to separate in-domain and OOD intents. Ren et al. (2019); Gangal et al. (2020) leverage likelihood ratios crafted by generative models to classify OOD samples. Wei et al. (2022) observe that the norm of the logits keeps increasing during training, leading to overconfident outputs, and propose LogitNorm as a solution to decoupling the output norm during training optimization. Moon et al. (2020) introduce a novel Correctness Ranking Loss function in order to regularize output probabilities to produce well-ranked confidence estimates. Other calibration techniques include "Top-label" calibration which is used to regularize the reported probability for the predicted class (Gupta and Ramdas, 2021).

# 3 SELFOOD: Self-Supervised OOD Detection

We present the problem statement, the motivation for our ranking formulation, and our method including the loss function and its optimization strategy.

## 3.1 Problem Statement

In this paper, we work on the OOD detection task with only in-distribution samples and no OOD samples as supervision. Specifically, given a labeled dataset $\mathcal{D}^{InD} = \{(x_1, y_1), (x_2, y_2), \ldots (x_n, y_n)\}$ sampled from a distribution space $(\mathcal{X}, \mathcal{C})$ where documents $x_i \in \mathcal{X}$ and labels $y_i \in \mathcal{C}$ as input, our goal is to train an OOD detector $\mathcal{M}$ that accurately distinguishes in-distribution documents $\mathcal{D}^{InD}$ and OOD documents $\mathcal{D}^{OOD} \notin (\mathcal{X}, \mathcal{C})$ without any

OOD documents required for training.

## 3.2 Motivation

Numerous neural text classifiers have been proposed, incorporating multiple hidden layers (Rosenblatt, 1957), convolutional layers (Kim, 2014), and various types of attention mechanisms (Devlin et al., 2019; Liu et al., 2019; Radford et al., 2019). All these models culminate in a softmax head, which produces probabilities corresponding to each class. These classifiers are generally trained with CE loss in an intra-document fashion i.e. each document is considered independently and the softmax score of the true label is maximized. Such training of neural text classifiers is known to increase the magnitude of logit vectors even when most training examples are correctly classified (Wei et al., 2022), making them poorly calibrated, producing unreasonably high probabilities even for incorrect predictions (Szegedy et al., 2013; Nguyen et al., 2015; Guo et al., 2017). This diminishes their ability to maintain the desired attribute of ordinal ranking for predictions based on confidence levels, wherein a prediction exhibiting a higher confidence value should be considered more likely to be accurate than one with a lower confidence value (Moon et al., 2020). Intuitively, a text classifier possessing such quality would be a perfect OOD detector.

## 3.3 OOD Detection as Inter-Document Intra-Label Ranking

In order to align with the aforementioned characteristic, we propose formulating the OOD detection as an inter-document, intra-label ranking problem. Specifically, given a set of in-distribution documents, we compare across documents but within the same label and train our model to generate higher probability score for documents belonging to the label than for documents not belonging to the label. We consider the same model architecture as any text classifier with a softmax head that generates scores corresponding to each label, however, we train it using our IDIL loss instead of CE loss. Our assumption is that an OOD document does not fall under any specific label in the in-distribution space. Hence, we anticipate that the trained model would produce lower scores for OOD documents compared to in-distribution documents. This distinction in scores is expected to facilitate easy separation between OOD and in-distribution documents.

**IDIL Loss** is a pairwise-ranking loss that enforces

desired ordinal ranking of confidence estimates. This loss function reaches its minimum value for a particular label when the probability of that label being the annotated label is greater than its probability when it is not the annotated label. Specifically, for documents $x_1, x_2 \in \mathcal{D}^{InD}$ and their corresponding annotated labels $y_1, y_2$ where $y_1 \neq y_2$, IDIL loss corresponding to label $y_1$ is mathematically computed as follows:

$$\mathcal{L}_{IDIL}(y_1|x_1, x_2) = SiLU(p(y_1|x_2) - p(y_1|x_1)) \tag{1}$$

where $SiLU(x) = x\sigma(x)$ is the Sigmoid Linear Unit (SiLU) function (Elfwing et al., 2018). To ensure stable training and enhance performance, we incorporate the SiLU function, a continuous variant of the Rectified Linear Unit (ReLU) (Hahnloser et al., 2000), in conjunction with the ranking loss. The SiLU function introduces smooth gradients around zero, effectively mitigating potential instability issues during training. We observe that this inclusion contributes to the overall stability in training and improved performance of the model as shown in Section 4.5. Note that, in contrast to CE loss, IDIL loss becomes zero once desired ranking is achieved, addressing the overconfidence issue.

### 3.4 Implementation

Ideally, the loss has to be computed over all possible pairs of documents for each model update. However, it is computationally expensive. Therefore, following (Toneva et al., 2019; Moon et al., 2020), we approximate the computation by considering only documents in each mini-batch. Specifically, we bucket the documents in the mini-batch based on their annotated label and pair each document in a bucket with all documents in other buckets and compute the loss. Mathematically, the loss for a mini-batch $b$ is computed as follows:

$$\mathcal{L} = \sum_{l \in \mathcal{C}} \sum_{x_1 \in b_l} \sum_{x_2 \in b_{\neg l}} \mathcal{L}_{IDIL}(l|x_1, x_2)$$

where $b_l$ denotes the set of training data points $x$ in this batch $b$ whose label are $l$, and $b_{\neg l}$ denotes the set of training data points $x$ in this batch $b$ whose label is *not* $l$.

In contrast to CE loss, where the optimization involves increasing the score corresponding to the correct label, we backpropagate gradients to decrease scores associated with incorrect labels. Specifically, during the backpropagation of gradients, we detach the gradients for the subtrahend

of the difference and exclusively propagate the gradients through the minuend. In Equation 1, for instance, we detach the gradients for $p(y_1|x_1)$ and solely backpropagate the gradients through $p(y_1|x_2)$. This detachment allows for a more controlled and selective gradient flow, aiding in the optimization process, and improvement in performance as shown in Section 4.5.

It is important to note that our optimization focuses solely on the inter-document ranking loss. Consequently, while the trained model would serve as a reliable OOD detector, it may not perform as effectively as a classifier.

## 4 Experiments

We evaluate our OOD detection method against state-of-the-art baselines with two classifiers on multiple datasets belonging to different domains.

### 4.1 Datasets

We evaluate our method and baselines on four publicly available English text classification datasets belonging to different domains. In particular, we consider the news topic classification dataset New York Times (NYT)[2], restaurant review sentiment classification dataset Yelp[3], and question-type classification datasets related to climate: Clima-Insurance+ (Clima) (Laud et al., 2023), and a general domain: TREC (Li and Roth, 2002; Hovy et al., 2001). The documents within the New York Times dataset are labeled with both coarse and fine-grained labels. For our training and testing process, we utilize fine-grained labels. The dataset statistics are provided in Table 1.

### 4.2 Compared Methods

- **Cross Entropy Loss (CE Loss)** trains a classifier using CE loss on in-distribution documents. The predicted probabilities from the classifier are used as confidence estimates for OOD detection.
- **Correctness Ranking Loss (CRL)** (Moon et al., 2020) is a regularization term added to the CE-

Table 1: Dataset statistics.

| Dataset | Domain | Criteria | # Docs | # labels |
|---------|--------|----------|--------|----------|
| **NYT** | News | Topic | 13081 | 26 |
| **Yelp** | Reviews | Sentiment | 70000 | 5 |
| **Clima** | Climate | Question Type | 17175 | 8 |
| **TREC** | General | Question Type | 5952 | 6 |

---

[2] http://developer.nytimes.com/
[3] https://www.yelp.com/dataset/

Table 2: OOD detection results with BERT & RoBERTa classifiers. Each experiment is repeated with three random seeds and the mean scores are reported. The false-positive-rate at 95% true-positive-rate (FPR95), minimum detection error over all thresholds (ERR), the area under the risk-coverage curve (AURC), and the area under the precision-recall curve (AUPR) using in-distribution samples as the positives are used as evaluation metrics.

| In-dist | OOD | Method | BERT | | | | RoBERTa | | | |
|---|---|---|---|---|---|---|---|---|---|---|
| | | | FPR95(↓) | ERR(↓) | AUROC(↑) | AUPR(↑) | FPR95(↓) | ERR(↓) | AUROC(↑) | AUPR(↑) |
| Yelp | NYT | LogitNorm | 82.0 | 33.2 | 69.4 | 57.9 | 84.4 | 33.9 | 66.6 | 55.0 |
| | | OneVsRest | 85.1 | 35.8 | 62.4 | 49.3 | 88.5 | 29.6 | 69.8 | 56.5 |
| | | RCL | **56.9** | 31.2 | 73.2 | 55.2 | 87.7 | 34.5 | 56.7 | 45.0 |
| | | Bayes Approx | 64.3 | 35.9 | 66.5 | 50.3 | 79.6 | 36.4 | 54.1 | 40.8 |
| | | CE Loss | 82.1 | 34.9 | 63.4 | 50.7 | 79.8 | 32.9 | 66.8 | 58.0 |
| | | CRL | 99.9 | 36.9 | 43.9 | 51.1 | 99.8 | 28.1 | 64.8 | 60.5 |
| | | SELFOOD | 63.2 | **19.6** | **79.4** | **82.7** | **69.5** | **21.6** | **72.6** | **78.8** |
| | Clima | LogitNorm | 79.2 | 25.5 | 72.5 | 54.8 | 82.6 | 25.4 | 70.7 | 54.0 |
| | | OneVsRest | 83.0 | 25.1 | 69.5 | 53.2 | 86.7 | 28.0 | 53.8 | 33.7 |
| | | RCL | 59.9 | 27.8 | 67.4 | 42.1 | 73.9 | 22.5 | 70.7 | 56.3 |
| | | Bayes Approx | 74.7 | 28.7 | 56.7 | 33.6 | **63.0** | 28.0 | 66.8 | 40.9 |
| | | CE Loss | 83.2 | 27.8 | 48.6 | 37.0 | 76.4 | 22.0 | **78.2** | 68.4 |
| | | CRL | 99.6 | 29.0 | 49.1 | 38.8 | 99.9 | 20.6 | 63.0 | 56.9 |
| | | SELFOOD | **17.6** | **4.8** | **97.9** | **96.4** | 65.6 | **16.6** | 73.2 | **77.2** |
| | TREC | LogitNorm | 68.2 | 27.0 | 79.8 | 82.1 | 60.8 | 23.0 | **83.9** | 84.4 |
| | | OneVsRest | 84.8 | 40.0 | 63.9 | 69.7 | 75.5 | 33.0 | 57.9 | 61.2 |
| | | RCL | 89.8 | 43.2 | 40.1 | 51.5 | 77.4 | 32.4 | 65.0 | 68.6 |
| | | Bayes Approx | 92.5 | 44.7 | 32.1 | 44.7 | **51.8** | 25.9 | 72.7 | 69.4 |
| | | CE Loss | 74.2 | 33.7 | 52.8 | 62.7 | 58.5 | 27.5 | 75.3 | 78.3 |
| | | CRL | 33.3 | 15.5 | 66.7 | 77.2 | 100.0 | 37.3 | 56.7 | 73.9 |
| | | SELFOOD | **0.0** | **0.0** | **100.0** | **100.0** | 55.1 | **20.6** | 80.2 | **90.3** |
| TREC | Yelp | LogitNorm | 32.3 | 0.3 | 95.5 | 78.1 | 15.2 | 0.5 | 95.7 | 44.7 |
| | | OneVsRest | 4.2 | 0.3 | 97.9 | 83.4 | 39.3 | 0.4 | 91.6 | 63.0 |
| | | RCL | 78.5 | 0.6 | 80.2 | 20.2 | 67.7 | 0.6 | 83.0 | 24.2 |
| | | Bayes Approx | 46.6 | 0.4 | 93.0 | 61.6 | 63.7 | 0.7 | 87.7 | 21.3 |
| | | CE Loss | 27.2 | 0.3 | 94.9 | 71.2 | 11.3 | 0.4 | 96.9 | 62.3 |
| | | CRL | 96.6 | 0.7 | 70.9 | 16.5 | 99.8 | 0.7 | 24.3 | 9.7 |
| | | SELFOOD | **0.0** | **0.0** | **100.0** | **100.0** | **0.0** | **0.0** | **100.0** | **100.0** |
| | NYT | LogitNorm | 9.3 | 1.0 | 97.7 | 91.2 | 10.9 | 2.3 | 97.2 | 77.3 |
| | | OneVsRest | 1.8 | 0.5 | 99.0 | 95.8 | 32.5 | 1.8 | 93.2 | 74.8 |
| | | RCL | 63.6 | 3.3 | 84.1 | 45.7 | 74.2 | 3.0 | 80.4 | 45.1 |
| | | Bayes Approx | 39.1 | 2.5 | 93.5 | 68.1 | 64.9 | 3.0 | 88.4 | 54.4 |
| | | CE Loss | 8.4 | 1.3 | 97.6 | 87.5 | 6.8 | 1.4 | 98.1 | 87.1 |
| | | CRL | 97.2 | 3.3 | 73.0 | 35.2 | 99.9 | 3.8 | 22.9 | 12.2 |
| | | SELFOOD | **0.0** | **0.0** | **100.0** | **100.0** | **0.0** | **0.0** | **100.0** | **100.0** |
| | Clima | LogitNorm | 11.3 | 0.8 | 97.6 | 88.5 | 10.4 | 1.6 | 97.3 | 75.9 |
| | | OneVsRest | 3.4 | 0.6 | 98.7 | 93.2 | 36.3 | 1.8 | 91.7 | 60.1 |
| | | RCL | 73.5 | 2.4 | 81.2 | 37.1 | 60.7 | 2.0 | 86.4 | 48.2 |
| | | Bayes Approx | 51.7 | 2.0 | 90.5 | 54.4 | 61.1 | 2.4 | 87.6 | 40.4 |
| | | CE Loss | 15.1 | 1.7 | 96.2 | 69.5 | 12.2 | 1.8 | 96.4 | 67.8 |
| | | CRL | 86.3 | 2.2 | 80.7 | 39.5 | 98.9 | 2.7 | 30.1 | 13.0 |
| | | SELFOOD | **0.0** | **0.0** | **100.0** | **100.0** | **0.0** | **0.0** | **100.0** | **100.0** |

loss to make class probabilities better confidence estimates. It estimates the true class probability to be proportional to the number of times a sample is classified correctly during training.

- **LogitNorm** (Wei et al., 2022) is a variant of CE loss that normalizes the logit vector to have a constant norm during training.
- **Bayesian Approximation (Bayes Approx)** (Wu et al., 2022b) calibrates distribution uncertainty for OOD detection using Monte-Carlo dropout.
- **Reassigned Contrastive Learning (RCL)** (Wu et al., 2022a) discriminates over-confident OOD samples using adaptive class-dependent local threshold mechanism.

- **OneVsRest** trains a one vs rest binary classifier per label. The probability scores of each label from its corresponding classifier are normalized and used as their respective confidence estimates.

### 4.3 Experimental Settings

We experiment with BERT (Devlin et al., 2019) and RoBERTa (Liu et al., 2019) as text classifiers. For SELFOOD, we train the classifier for 5 epochs with a batch size of 16 using an AdamW optimizer. We use a learning rate of 5e-5 using a linear scheduler with no warm-up steps. For all baselines, we train the classifier for the same number of steps.

In our evaluation, for each dataset as in-

distribution, we treat all other datasets as OOD and compute the performance. Our evaluation follows a standard approach for each in-distribution dataset where we begin by splitting the in-distribution dataset into three subsets: 80% for train, 10% for val, and 10% for test. The model is trained using the train split, and its performance is evaluated on both the test split of the in-distribution dataset and the entire OOD dataset.

**Evaluation Metrics.** We utilize evaluation metrics from (Hendrycks and Gimpel, 2017; DeVries and Taylor, 2018; Moon et al., 2020) such as the false positive rate at 95% true positive rate (FPR95), minimum detection error over all thresholds (ERR), the area under the risk-coverage curve (AURC), and the area under the precision-recall curve (AUPR) using in-distribution samples as the positives.

### 4.4 Results

We summarize the evaluation results with BERT and RoBERTa as classifiers on Yelp, TREC as in-distribution in Table 2 and NYT, Clima as in-distribution in Appendix A.1. All experiments are run on three random seeds and the mean performance scores are reported. As shown in Table 2, we observe that SELFOOD performs better than the baselines on most of the in-distribution, OOD dataset pairs for both classifiers. A low FPR95 value indicates that the top-95% confident in-distribution samples, selected based on their probability scores, predominantly rank higher than a majority of OOD samples and SELFOOD achieves improvements of up to 82 points in FPR95 with Yelp as in-distribution and Clima as OOD datasets when compared to CRL with BERT classifier. SELFOOD also exhibits substantial improvements of up to 33 points in Detection Error, 48 points in AUROC, and 58 points in AUPR when compared to CE-Loss with BERT classifier. SELFOOD achieves a perfect OOD detection score for some settings such as TREC as in-distribution dataset for both BERT and RoBERTa classifiers. These results highlight the effectiveness of our ranking formulation with self-supervised training using IDIL loss.

### 4.5 Ablation Study

To understand the impact of each component in our IDIL loss design and implementation, we compare our method with four ablated versions with BERT classifier in Table 3: (1) SELFOOD + *Grad Sub* represents our method with backpropagating gradi-

Table 3: Ablation Study.

| Method | FPR95 | ERR | AUROC | AUPR |
|---|---|---|---|---|
| | | In Dist: Clima OOD: NYT | | |
| SELFOOD | 17.2 | **2.6** | **97.8** | **94.5** |
| + Grad Sub | 67.9 | 8.9 | 82.4 | 61.1 |
| + Grad Sub & Min | 98.4 | 10.7 | 49.2 | 32.3 |
| + Intra-Doc | 92.7 | 10.6 | 67.3 | 41.6 |
| - SILu | **14.6** | 3.9 | 96.7 | 89.3 |
| | | In Dist: NYT OOD: Clima | | |
| SELFOOD | **10.5** | **1.7** | **98.4** | **90.5** |
| + Grad Sub only | 83.5 | 5.1 | 77.4 | 44.8 |
| + Grad Sub & Min | 96.5 | 5.2 | 60.3 | 36.0 |
| + Intra-Doc | 48.3 | 2.6 | 93.2 | 80.2 |
| - SiLU | 19.0 | 2.3 | 97.4 | 86.0 |

ents through the subtrahend instead of the minuend, (2) SELFOOD + *Grad Sub & Min* represents our method with gradients backpropagating through both minuend and subtrahend, (3) SELFOOD + *Intra-Doc* considers intra-document comparison similar to CE loss, in addition to inter-document intra-label comparison for loss computation, and (4) SELFOOD- *SiLU* excludes SiLU function from the IDIL loss formulation. We also present the performance of SELFOOD for reference. SELFOOD performs better than SELFOOD + *Grad Sub & Min* demonstrating that backpropagating through one part of the difference is more beneficial than both, and the comparison between SELFOOD and SELFOOD + *Grad Sub* indicates that backpropagating through minuend is better than subtrahend. We observe that incorporating intra-document comparison into loss formulation leads to a decrement in the ranking ability of the model. Finally, we observe that removing the SiLU function from the IDIL loss leads to a decrease in most of the metrics.

## 5 Analysis & Case Studies

In this section, we present a comprehensive analysis of our proposed method from different perspectives to understand its effectiveness.

### 5.1 CE Loss vs SELFOOD: Training Trajectory Analysis

Mathematically, the model trained using CE loss is a solution to our IDIL loss optimization. However, our empirical results reveal a significant disparity in OOD detection performance between SELFOOD and model trained on CE loss. To gain a deeper understanding, we plot the CE Loss and IDIL losses for models optimized with IDIL and CE loss, as shown in Figure 3(a). We consider Clima as the in-distribution dataset and BERT as the classifier.

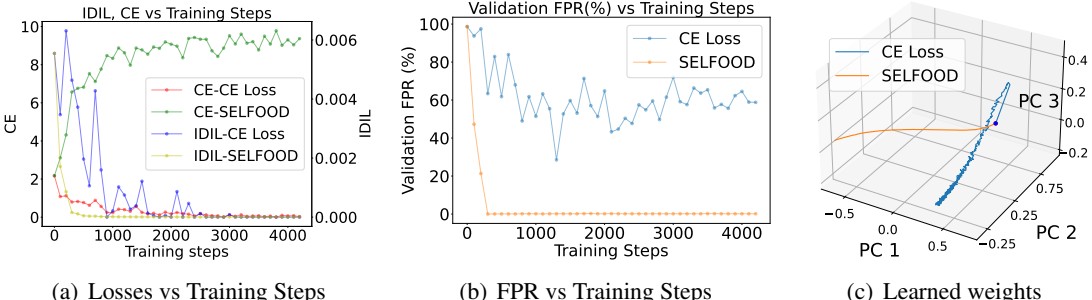

(a) Losses vs Training Steps     (b) FPR vs Training Steps     (c) Learned weights

Figure 3: To understand training trajectories of CE Loss and SELFOOD, we plot their respective losses, validation FPR, and learned weights while training. We observe IDIL loss minimization is more correlated with performance and the learned weights of CE-Loss and SELFOOD show divergent paths since the beginning.

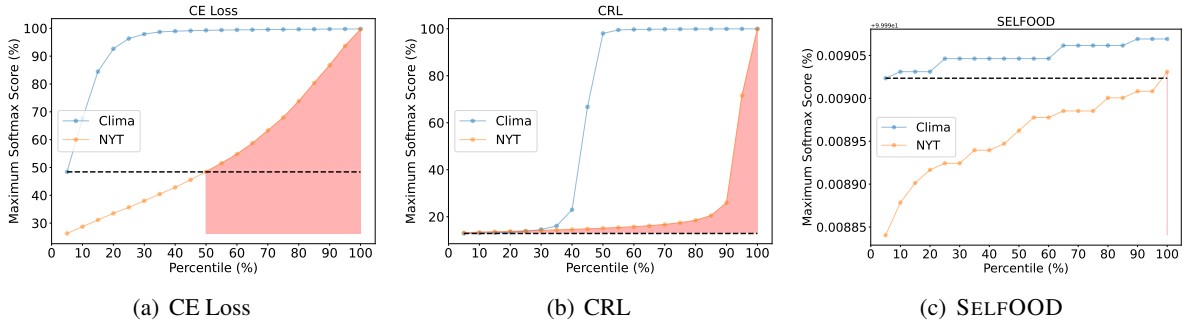

(a) CE Loss           (b) CRL           (c) SELFOOD

Figure 4: Maximum softmax score vs percentile in-distribution & OOD data associated with that score. By using the least in-distribution maximum softmax score as the threshold (dotted line) for OOD classification, CE loss considers more than 50% and CRL considers almost 100% of the OOD data as in-distribution (red region). However, in the case of SELFOOD, we observe a clear margin in maximum softmax scores that separate OOD and in-distribution.

Additionally, we plot the validation FPR95 using Yelp as the OOD dataset to analyze the correlation between loss minimization and performance in Figure 3(b). Figure 3(a) illustrates that while minimizing CE loss leads to fluctuations in the IDIL loss, ultimately reaching zero, Figure 3(b) shows that the performance is more closely associated with explicit IDIL minimization. Specifically, minimizing the IDIL loss results in improved FPR95. The findings indicate that the minimization of CE loss and IDIL loss follows distinct trajectories. To validate this hypothesis, we conduct an experiment using a toy 3-dimensional dataset comprising five isometric Gaussian blobs and train a logistic regression classifier initialized with same weights, minimizing IDIL and CE-loss. We flatten the weight matrix and plot the top-3 principal components. As shown in Figure 3(c), the learned weights of CE-Loss and SELFOOD progress in different directions, thereby confirming our hypothesis.

## 5.2 Maximum softmax score Analysis

We plot the maximum softmax score with the percentile in-distribution and OOD data associated with that score for CE loss, CRL, and SELFOOD

with BERT classifier on Clima as in-distribution and NYT as OOD datasets in Figure 4. When comparing the maximum softmax score with the percentile data associated with that score, interesting observations can be made regarding its distribution. Specifically, when using the least in-distribution maximum softmax score as the threshold for OOD classification, we find that the CE loss considers over 50% of the OOD data and CRL considers almost 100% of the OOD data as in-distribution. However, in the case of SELFOOD, we observe a clear margin in the maximum softmax scores that effectively separates OOD and in-distribution data. This suggests that the threshold needs to be carefully tuned for CE loss and CRL, requiring more effort and annotated data, whereas in the case of SELFOOD, we do not require such tuning. This demonstrates that SELFOOD is capable of accurately classifying between OOD and in-distribution samples based on their respective maximum softmax scores, resulting in superior performance.

## 5.3 Batch size Analysis

We compare documents within the mini-batch when computing the loss. Consequently, the num-

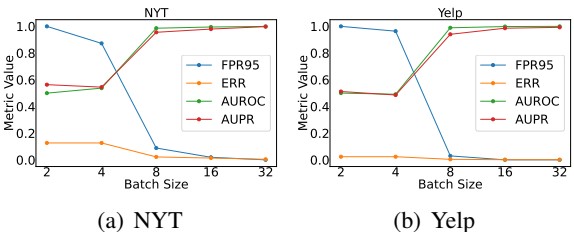

|  | (a) NYT | (b) Yelp |

Figure 5: Performance of SELFOOD w.r.t. batch size. We consider BERT classifier, Clima as in-distribution, and NYT, Yelp as OOD datasets. We observe an increase in performance with an increase in batch size.

ber of document pairs over which the loss is computed depends on the batch size used during training. To investigate the impact of batch size on performance, we vary the batch size and evaluate the corresponding performance in Figure 5 with BERT classifier. We consider Clima as in-distribution, and NYT, Yelp as OOD datasets. As the batch size increases, the performance of the model also improves, until a batch size of 16, where the performance reaches a plateau. This observation aligns with our intuition that a larger batch size allows for more in-batch document pairs to be compared, leading to more accurate loss computation. Based on these findings, we recommend utilizing a batch size of 16 or higher to achieve optimal performance.

## 5.4 Fine-grained OOD Detection

To explore the limits of SELFOOD, we consider in-distribution and OOD datasets from the same domain. The objective is to assess its ability to differentiate between samples that are closely related but still OOD, which is a challenging task. Specifically, we consider the news domain and choose NYT as the in-distribution, and AGNews (Zhang et al., 2015), 20News[4] as OOD datasets and train BERT classifier. As shown in Table 4, SELFOOD performs significantly better than the baselines on most of the metrics. Moreover, it also achieves near-perfect scores for the AGNews, highlighting its ability to accurately identify OOD samples even when they belong to the same domain as the in-distribution dataset.

## 5.5 Classification performance of SELFOOD

The BERT classifier trained with CE-Loss demonstrates exceptional classification performance, achieving accuracy scores of 97%, 85%, and 97% on NYT, Clima, and TREC datasets respectively. In contrast, SELFOOD achieves only 0.3%, 9.5%,

---

[4] http://qwone.com/~jason/20Newsgroups/

Table 4: Fine-grained OOD detection results with NYT as the in-distribution dataset with BERT classifier. We choose two datasets AGNews, 20News from the news domain, the same as NYT, and consider them as OOD datasets. The results show that SELFOOD can accurately detect OOD samples within the same domain.

| OOD | Method | FPR95 | ERR | AUROC | AUPR |
|---|---|---|---|---|---|
| AGNews | LogitNorm | 45.9 | 0.9 | 86.7 | 15.8 |
|  | Bayes Approx | 57.5 | 0.9 | 87.7 | 23.9 |
|  | CE-Loss | 45.5 | 0.9 | 86.5 | 16.6 |
|  | CRL | 77.2 | 0.9 | 80.6 | 9.5 |
|  | SELFOOD | **0.0** | **0.1** | **100.0** | **99.4** |
| 20News | LogitNorm | **24.5** | 6.2 | 91.7 | 39.2 |
|  | Bayes Approx | 35.0 | **4.3** | 93.7 | **66.6** |
|  | CE-Loss | 25.4 | 6.2 | 91.9 | 39.6 |
|  | CRL | 59.9 | 6.4 | 87.4 | 36.9 |
|  | SELFOOD | 28.2 | 4.9 | **94.2** | 61.5 |

and 1.8%. This highlights that while SELFOOD serves as a reliable OOD detector, its performance as a text classifier is subpar. This observation can be attributed to our IDIL loss formulation, which focuses on comparing confidence levels across documents for each label rather than across labels for each document. As a result, the IDIL loss primarily promotes the ordinal ranking of confidence levels across documents, which enhances the model's OOD detection capabilities. However, this emphasis on inter-document ranking comes at the expense of inter-label ranking, resulting in limited classification capabilities. Moreover, when we introduce the intra-document comparison to the IDIL loss, as discussed in Section 4.5, we observe a decline in the model's ranking ability. Podolskiy et al. (2021) use mahalanobis distance for OOD detection and observe that although mahalanobis distance alongside transformer-based models is effective, such methods are sensitive to the geometric features in the embedding space and can be spoilt if the embedder is used for classification and is overfit. This matches with our observation, upon adding intra-document comparison to the loss, the OOD detection performance drops. This further supports the notion that balancing the inter-document and intra-document comparisons is crucial for achieving optimal performance in both OOD detection and text classification tasks.

## 5.6 SELFOOD + Mahalanobis distance

Mahalanobis distance-based estimation of OOD scores is an effective post-processing method used on trained OOD classifiers (Leys et al., 2018; Xu et al., 2020). We investigate whether this post-processing further improves the performance of our method. OOD scores are estimated as the dis-

Table 5: OOD detection results with Mahalanobis Distance post-processing technique. We choose two in-distribution, OOD pairs with BERT classifier. The results show that SELFOOD's OOD detection capabilities are enhanced with the post-processing technique.

| InD | OOD | Method | FPR95 | ERR | AUROC | AUPR |
|------|------|-----------------|-------|------|-------|------|
| Clima | Yelp | SELFOOD | 14.7 | 0.6 | 98.0 | 90.8 |
|      |      | SELFOOD + Maha | **0.8** | **0.2** | **99.2** | **97.2** |
| Yelp | NYT | SELFOOD | 63.2 | 19.6 | 79.4 | 82.7 |
|      |      | SELFOOD + Maha | **12.4** | **4.4** | **97.4** | **97.5** |

tance between a test data point and the distribution of in-distribution samples using the Mahalanobis distance metric. Following (Xu et al., 2020), we consider the intermediate layer encodings of the OOD classifier trained on in-distribution samples as its representative distribution. We apply this post-processing on top of SELFOOD and experiment with two in-distribution, OOD pairs with BERT classifier. As shown in Table 5, we observe further improvement in performance, demonstrating the quality of learned representative distribution using SELFOOD.

### 5.7 Qualitative Analysis of SELFOOD

We present a few examples with Yelp as in-distribution and TREC as OOD where SELFOOD accurately distinguishes and CE-loss fails to distinguish between in-distribution and OOD samples in Table 6. Although the first example has food-related words like *lunch, cork* and sentiment-indicative words such as *contemptible scoundrel*, SELFOOD accurately identifies it as OOD and CE loss model fails to do so. Examples 3 and 4 have questions that revolve around eateries. CE-Loss model confuses and predicts them as OOD as TREC dataset contains only questions. However, SELFOOD was able to rank them higher and accurately identify them as in-distribution.

### 6 Conclusion

In this paper, we present SELFOOD, a novel framework for OOD detection that leverages only in-distribution samples as supervision. Building upon the insight that OOD samples typically do not belong to any in-distribution labels, we formulate the OOD detection problem as an inter-document intra-label ranking task. To address this challenge, we propose IDIL loss which guides the training process. Through extensive experiments on multiple datasets, we demonstrate the effectiveness of SELFOOD on OOD detection task. However, we

Table 6: Qualitative analysis of CE-Loss vs SELFOOD with Yelp as in-distribution and TREC as OOD datasets. Wrong predictions are in Red and correct predictions are in Green.

| Text | CE-Loss | SELFOOD |
|------|---------|---------|
| Who said "What contemptible scoundrel stole the cork from my lunch?" | InD | OOD |
| Who said "Give me liberty or give me death"? | InD | OOD |
| Its a tequila house - wouldn't you want your staff to be knowledgeable about tequila? Call me silly. So Gregg and I are wondering Bourbon Street, looking for the next "place"... | OOD | InD |
| the pitch reads: "what's worse than mean girl cheerleaders? how about resurrected mean-girl cheerleaders with supernatural powers?" And if this place could be farther away from my house, it'd be in Camden... Oh, the Vomitorium, everytime i grace the hipster-stained .. | OOD | InD |

also acknowledge that it comes at the expense of text classification performance. Future research can focus on developing techniques that effectively balance inter-document and intra-document comparisons, enabling improved performance in both OOD detection and text classification tasks.

### 7 Limitations

As mentioned in the previous sections, the OOD detection performance of our method comes at the cost of classification performance. This limitation needs to be addressed.

### 8 Ethics Statement

This paper proposes a self-supervised OOD detection method. The aim of the paper is to minimize the human effort in annotating OOD documents and using only in-distribution documents for OOD detection. OOD detection is helpful in detection potential harmful content. Hence, we do not anticipate any major ethical concerns.

### 9 Acknowledgments

We thank the anonymous reviewers and our colleagues from Shang Data Lab, especially Bill Hogan and Zihan Wang for their helpful feedback. Our work is sponsored in part by NSF CAREER Award 2239440, NSF Proto-OKN Award 2333790, NIH Bridge2AI Center Program under award 1U54HG012510-01, Cisco-UCSD Sponsored Research Project, as well as generous gifts from Google, Adobe, and Teradata. Any opinions, findings, and conclusions or recommendations expressed herein are those of the authors and should

not be interpreted as necessarily representing the views, either expressed or implied, of the U.S. Government. The U.S. Government is authorized to reproduce and distribute reprints for government purposes not withstanding any copyright annotation hereon.

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

# A Appendix

## A.1 Results with NYT & Clima as in-distribution datasets

We present experimental results on Clima and NYT as in-distribution datasets with BERT and RoBERTa classifiers in Table 7.

## A.2 Experimental Settings

**Training trajectory analysis experiment.** We compare two experiments showcasing empirical evidence on how the trajectory of CE-loss models and IDI loss models differ. In 3(b) we plot the change in validation FPR on BERT-classifiers optimized on CE-loss and IDIL loss. Here, we calculate the FPR across training steps on a validation dataset consisting of 500 in-distribution and OOD data points each to compare the training trajectories of the two models.

We generate the toy dataset by creating five isometric Gaussian blobs of 3-dimensional data points, with each cluster containing 1000 samples. The centers of all the clusters are bounded by the range $[-10.0, 10.0]$ and the cluster standard deviation is set to 2.0 to account for limited overlap between cluster datapoints. We consider four clusters as being in-distribution and the remaining cluster as OOD. We train a logistic regression classifier on the in-domain datapoints initialized with the same weights and minimizing IDIL and CE-loss. The model is optimized using *Stochastic Gradient Descent* with a learning rate of 0.01 for 300 epochs. The weights are visualized at the begining of each epoch. In order to facilitate visualization, the weight matrices of both approaches of dimension are flattened and *Principal Component Analysis (PCA)* is applied to reduce the dimensionality of the weight matrix. We plot the top-3 principal components after eacch epoch to conclude that the learned weights of CE-loss and SELFOOD progress in a different maner.

| In-dist | OOD | Method | BERT | | | | RoBERTa | | | |
|---|---|---|---|---|---|---|---|---|---|---|
| | | | FPR95(↓) | ERR(↓) | AUROC(↑) | AUPR(↑) | FPR95(↓) | ERR(↓) | AUROC(↑) | AUPR(↑) |
| NYT | Clima | LogitNorm | 20.6 | **1.0** | 96.8 | 92.3 | 18.4 | **1.0** | 97.3 | 93.6 |
| | | OneVsRest | 84.7 | 1.7 | 87.3 | 82.6 | 92.7 | 1.9 | 91.1 | 85.1 |
| | | RCL | 46.9 | 1.5 | 94.4 | 87.5 | 86.7 | 2.3 | 86.7 | 76.6 |
| | | Bayes Approx | 40.6 | 1.2 | 95.4 | 90.0 | 80.0 | 2.4 | 88.4 | 76.5 |
| | | CE Loss | 22.1 | **1.0** | 96.9 | **92.5** | 15.7 | **1.0** | 97.3 | 92.8 |
| | | CRL | 32.3 | 1.7 | 95.4 | 86.5 | 79.3 | 3.2 | 79.8 | 67.0 |
| | | SELFOOD | **10.5** | 1.6 | **98.4** | 90.5 | **4.6** | **1.0** | **99.3** | **95.4** |
| | TREC | LogitNorm | 13.3 | 5.5 | 97.2 | 87.9 | 17.5 | 6.4 | 96.1 | 84.1 |
| | | OneVsRest | 96.0 | 6.2 | 84.0 | 79.4 | 97.2 | 6.3 | 90.5 | 82.8 |
| | | RCL | 62.9 | 6.1 | 92.2 | 85.8 | 73.9 | 8.8 | 88.4 | 73.1 |
| | | Bayes Approx | 38.8 | 4.5 | 94.8 | 89.9 | 67.5 | 8.3 | 88.7 | 74.7 |
| | | CE Loss | 14.5 | 5.8 | 97.0 | 87.1 | 21.9 | 6.7 | 95.5 | 83.5 |
| | | CRL | 67.9 | 6.5 | 89.9 | 79.1 | 85.6 | 9.6 | 71.4 | 61.4 |
| | | SELFOOD | **0.0** | **0.0** | **100.0** | **100.0** | **0.0** | **0.0** | **100.0** | **100.0** |
| | Yelp | LogitNorm | **1.7** | 0.6 | **99.3** | 87.9 | 1.2 | 0.6 | 98.5 | 87.9 |
| | | OneVsRest | 15.1 | 0.7 | 97.4 | 78.1 | 8.7 | 0.6 | 97.6 | 78.8 |
| | | RCL | 45.1 | 0.5 | 94.9 | 84.7 | 76.7 | 1.0 | 90.2 | 60.0 |
| | | Bayes Approx | 28.7 | **0.4** | 96.4 | 86.8 | 83.0 | 1.1 | 87.7 | 56.6 |
| | | CE Loss | 3.4 | 0.7 | 99.2 | 82.3 | 1.8 | 0.6 | 99.3 | 85.4 |
| | | CRL | 69.5 | 0.5 | 89.3 | 77.8 | 73.3 | 1.0 | 83.8 | 58.3 |
| | | SELFOOD | 22.5 | 1.1 | 95.7 | 57.8 | **0.2** | **0.2** | **99.9** | **97.5** |
| Clima | TREC | LogitNorm | 37.8 | 5.5 | 94.6 | 91.9 | 60.5 | **7.0** | **91.6** | **88.0** |
| | | OneVsRest | 25.5 | 6.8 | 95.5 | 91.8 | 54.3 | 15.5 | 84.5 | 67.7 |
| | | RCL | 80.1 | 18.4 | 75.7 | 55.3 | 91.2 | 18.4 | 66.6 | 47.7 |
| | | Bayes Approx | 83.7 | 15.8 | 77.5 | 63.8 | 65.7 | 13.9 | 83.6 | 70.7 |
| | | CE Loss | 34.9 | 5.6 | 94.9 | 92.2 | 60.2 | 9.1 | 90.4 | 83.8 |
| | | CRL | 83.0 | 12.7 | 82.0 | 72.2 | 60.9 | 12.6 | 79.2 | 63.2 |
| | | SELFOOD | **0.3** | **1.1** | **99.4** | **99.4** | 38.9 | 8.8 | 90.1 | 84.7 |
| | Yelp | LogitNorm | 57.7 | 0.8 | 93.0 | 81.3 | 54.7 | **1.2** | **92.6** | **72.3** |
| | | OneVsRest | 43.7 | 1.0 | 92.7 | 72.5 | 58.8 | 1.8 | 73.5 | 34.7 |
| | | RCL | 91.0 | 2.3 | 60.7 | 12.2 | 98.0 | 2.3 | 56.4 | 12.7 |
| | | Bayes Approx | 81.4 | 2.2 | 76.5 | 22.7 | 96.8 | 2.2 | 52.3 | 16.3 |
| | | CE Loss | 46.3 | 0.9 | 93.8 | 80.1 | 79.5 | 1.9 | 86.5 | 47.2 |
| | | CRL | 98.7 | 1.4 | 67.3 | 53.3 | 72.9 | 2.1 | 50.9 | 19.0 |
| | | SELFOOD | **14.7** | **0.6** | **98.0** | **90.8** | **37.6** | 1.5 | 72.3 | 54.9 |
| | NYT | LogitNorm | 37.6 | 4.2 | 94.4 | 87.3 | 46.1 | **4.5** | **93.1** | **85.4** |
| | | OneVsRest | 34.6 | 4.8 | 94.3 | 85.3 | **40.6** | 9.4 | 83.1 | 51.3 |
| | | RCL | 90.3 | 12.1 | 64.6 | 28.8 | 97.9 | 12.4 | 51.5 | 21.2 |
| | | Bayes Approx | 87.4 | 11.2 | 70.4 | 38.3 | 96.8 | 11.3 | 55.8 | 29.2 |
| | | CE Loss | 44.9 | 4.8 | 93.2 | 84.4 | 62.5 | 6.7 | 89.6 | 75.2 |
| | | CRL | 85.6 | 7.6 | 78.0 | 62.5 | 61.3 | 9.4 | 61.8 | 39.9 |
| | | SELFOOD | **17.2** | **2.6** | **97.8** | **94.5** | 55.3 | 6.7 | 68.6 | 61.5 |

Table 7: OOD detection results with BERT & RoBERTa classifiers. Each experiment is repeated with three random seeds and the mean scores are reported. The false-positive-rate at 95% true-positive-rate (FPR95), minimum detection error over all thresholds (ERR), the area under the risk-coverage curve (AURC), and the area under the precision-recall curve (AUPR) using in-distribution samples as the positives are used as evaluation metrics.