# OpenReview forum: "SELFOOD: Self-Supervised Out-Of-Distribution Detection via Learning to Rank"
_EMNLP/2023/Conference — EMNLP 2023 Findings_

### Official Review · Reviewer_bvkd · 2023-08-03

**Soundness:** 3

**Excitement:**

2: Mediocre: This paper makes marginal contributions (vs non-contemporaneous work), so I would rather not see it in the conference.

**Paper Topic And Main Contributions:**

This paper introduces SELFOOD, a self-supervised out-of-distribution (OOD) detection method that addresses the annotation bottleneck and achieves effective OOD detection through inter-document intra-label ranking. The authors simplify the pair-wise computation by only considering documents in each mini-batch. SELFOOD requires only in-distribution samples as supervision and does not rely on any external datasets or pre-trained models. The proposed method is evaluated on several benchmark datasets and outperforms state-of-the-art supervised and unsupervised methods. Additionally, SELFOOD is shown to be effective in both coarse and fine-grained classification settings. Overall, SELFOOD presents a promising approach to OOD detection that can significantly reduce the need for labeled OOD data.

**Questions For The Authors:**

1. In the paper, do you assume that OOD samples do not belong to any in-distribution labels?
2. Have you done any study on replacing LLM objective with the proposed loss?

**Reasons To Accept:**

The paper introduces a promising approach to OOD detection that can significantly reduce the need for labeled OOD data. This could have practical implications for real-world NLP applications where labeled OOD data is often scarce or expensive to obtain.

**Reasons To Reject:**

The paper is well-structured, but the contribution isn't sufficient for a long paper. An interesting follow-up study would be exploring how the proposed loss can be utilized in LLM training - for instance, improving the current LLM training objective using this proposed idea.

**Reproducibility:**

4: Could mostly reproduce the results, but there may be some variation because of sample variance or minor variations in their interpretation of the protocol or method.

**Reviewer Confidence:**

3: Pretty sure, but there's a chance I missed something. Although I have a good feel for this area in general, I did not carefully check the paper's details, e.g., the math, experimental design, or novelty.

---

> ### Author Rebuttal · Authors · 2023-08-29
>
> __Contributions__
> In this paper, we present SELFOOD, a novel framework for OOD detection that leverages only in-distribution samples as supervision. Building upon the insight that OOD samples typically do not belong to any in-distribution labels, we formulate the OOD detection problem as an inter-document intra-label ranking task. To address this challenge, we propose IDIL loss which guides the training process. Through extensive experiments on multiple datasets, we demonstrate the effectiveness of SELFOOD on OOD detection task. We believe our paper has significant contributions and appreciate it if you reconsider your comment.
>
> __Assumption__
> Yes. As stated in lines 217-218, we assume that an OOD document does not fall under any specific label in the in-distribution space.
>
> __LLM with IDIL loss__
> We cannot adapt SELFOOD to language modeling because our method is only for OOD detection and not for classification whereas language modeling invovles token classification at each step.

---

### Official Review · Reviewer_At17 · 2023-08-13

**Soundness:** 2

**Excitement:**

3: Ambivalent: It has merits (e.g., it reports state-of-the-art results, the idea is nice), but there are key weaknesses (e.g., it describes incremental work), and it can significantly benefit from another round of revision. However, I won't object to accepting it if my co-reviewers champion it.

**Paper Topic And Main Contributions:**

The paper focus on self-supervised out-of-distribution detection, and proposes a novel method named SELFOOD, which only utilizes annotated in-distribution data as supervision. The main contribution is formulating the task as an inter-document intra-label (IDIL) ranking problem, and proposing the IDIL loss for backpropagation. The proposed method outperforms most of the baseline methods on various datasets.

**Questions For The Authors:**

In Table 2, it’s observed the proposed method outperforms baselines by a large margin. Are all methods using the same backbone model? Is the comparison fair for all the baselines and the designed approach?

**Reasons To Accept:**

The task of self-supervised OOD detection is valuable for reducing the annotation cost and is under-investigated.

The idea of inter-document intra-label ranking for calibration is novel and reasonable.

The proposed method is simple and effective on the OOD detection task.

**Reasons To Reject:**

It’s claimed that training with the proposed IDIL loss results in a more calibrated model, are there any experiments results to support the statement? Additional empirical results or theoretical analysis besides OOD detection metrics (FPR 95, AUROC) is suggested to support the statement and show the effectiveness of the proposed method.

It seems that the results on the ACC@FPRn metric are not reported in the paper. As mentioned, the detection performance of the proposed method comes at the cost of classification performance, it’s unclear about the ID accuracy compared to baselines. It’s strongly advised to report ACC@FPRn in the main text, especially ACC@FPR0.

**Reproducibility:**

3: Could reproduce the results with some difficulty. The settings of parameters are underspecified or subjectively determined; the training/evaluation data are not widely available.

**Reviewer Confidence:**

3: Pretty sure, but there's a chance I missed something. Although I have a good feel for this area in general, I did not carefully check the paper's details, e.g., the math, experimental design, or novelty.

---

> ### Author Rebuttal · Authors · 2023-08-29
>
> __Calibrated Model__
> We wish to emphasize that our method does not yield a calibrated model and we never stated so in the paper. Instead, it yields a model that conforms to the desired ordinal ranking, wherein the predicted probabilities align with the intended hierarchy, i.e., the probability of the correct label surpasses that of incorrect labels.  An ideally calibrated model produces probabilities whose magnitudes accurately represent their associated confidences, resulting in the aforementioned ranking. However, the probability scores generated by the model trained with the IDIL loss function do not reliably signify its level of confidence. Given that accurately ranked probabilities are sufficient for precise OOD detection, our approach delivers a high-quality OOD classifier.
>
> __Accuracy Results__
> We discuss and analyse the classification performance of SELFOOD with BERT classifier on NYT, Clima, and TREC datasets in section 5.5. SELFOOD performs poorly compared to other methods in classification. This is due to our IDIL loss formulation which promotes the ordinal ranking of confidence levels. It compares confidence levels across documents for each label rather than across labels for each document, which is required for high accuracy. Therefore, we emphasize in lines 90-94 that our method is specifically designed to improve the performance of OOD detection rather than classification accuracy.
>
> __Backbone Model & Fair Comparison__
> As stated in lines 331-337, we consider two backbone models BERT & RoBERTa for all baselines and similar experimental settings such as epochs and learning rate for a fair comparison.

---

### Official Review · Reviewer_6CvH · 2023-08-13

**Typos Grammar Style And Presentation Improvements:** NA
**Soundness:** 3

**Excitement:**

4: Strong: This paper deepens the understanding of some phenomenon or lowers the barriers to an existing research direction.

**Missing References:**

NA

**Paper Topic And Main Contributions:**

This paper proposes SELFOOD, a self-supervised OOD detection framework that requires only in-distribution samples as supervision. SELFOOD formulates OOD detection as an inter-document intra-label ranking problem and optimizes it using the proposed IDIL ranking loss. Compared with traditional CE loss where optimizing the probability of the correct label often leads to overconfidence, the IDIL ranking loss trains the model to rank the considered label probability score in documents belonging to the label higher than those not belonging to it. The paper evaluates SELFOOD and baselines on four text classification datasets: NYT, Yelp, Clima, and TREC.  Extensive experiments and analyses demonstrate the effectiveness of the proposed method.

**Questions For The Authors:**

Could the authors justify why SELFOOD is only evaluated on the task of text classification?

**Reasons To Accept:**

- The proposed SELFOOD method is well-motivated and sound.

- The paper conducted a comprehensive comparison with SOTA baselines.
- The performance improvements brought by SELFOOD are quite significant on all four datasets.
- The paper is well-written and easy to follow.

**Reasons To Reject:**

- The major concern is that while the paper claimed to propose a general self-supervised method for OOD detection, it was only evaluated on the task of text classification. There are other common tasks and datasets for OOD detection, such as intent classification, sentiment analysis, and topic prediction. However, the paper did not evaluate on them.
- Intent classification for dialogue systems is the most common application for OOD detection, and there are many datasets such as CLINIC150, STAR, ROSTD, SNIPS, ATIS for intent detection are proposed. Without evaluating on them, it is a bit hard to demonstrate the generalizability of SELFOOD.

**Reproducibility:**

4: Could mostly reproduce the results, but there may be some variation because of sample variance or minor variations in their interpretation of the protocol or method.

**Reviewer Confidence:**

3: Pretty sure, but there's a chance I missed something. Although I have a good feel for this area in general, I did not carefully check the paper's details, e.g., the math, experimental design, or novelty.

---

> ### Author Rebuttal · Authors · 2023-08-29
>
> __Intent classification datasets__
> Thanks for the suggestion! We experiment with SNIPS as in-distribution dataset and Clinic 150 as OOD dataset with BERT classifier. The minimum detection error of CE-loss model is 2.0 and SELFOOD is 1.76. The results demonstrate improvement with SELFOOD over CE-loss. We will add more experiment results on intent classification datasets in the final version.

---

### Official Review · Reviewer_MNGK · 2023-08-13

**Soundness:** 3

**Excitement:**

4: Strong: This paper deepens the understanding of some phenomenon or lowers the barriers to an existing research direction.

**Paper Topic And Main Contributions:**

This paper aims at the out-of-distribution (OOD) detection issue for classification. They present SelfOOD to address this annotation bottleneck and require only in-distribution labels. They train the target classifier to rank softmax scores instead of the standard cross-entropy loss. This ranking score then can help decrease probabilities associated with incorrect labels.

**Questions For The Authors:**

+ Apart from discrete classification, is there any way to adopt SelfODD for continuous labels? This can make SelfODD more general in various usages.


**Reasons To Accept:**

+ This paper is well-written and easy to follow.
+ The proposed ranking loss has the potential to be a new loss function for general classification tasks.
+ The proposed SelfODD shows significant improvements on both in- and out-domain classifications across various datasets.


**Reasons To Reject:**

+ Though having promising results, there is a lack of motivation or theoretical discussion on why this ranking can be that effective. From my understanding, the standard CE loss will also decrease the incorrect probabilities (though not that actively).
+ It can be better to present some qualitative examples and demonstrate how SelfODD achieves the correct prediction compared to the baseline.


**Reproducibility:**

4: Could mostly reproduce the results, but there may be some variation because of sample variance or minor variations in their interpretation of the protocol or method.

**Reviewer Confidence:**

4: Quite sure. I tried to check the important points carefully. It's unlikely, though conceivable, that I missed something that should affect my ratings.

---

> ### Author Rebuttal · Authors · 2023-08-29
>
> __Effectiveness of IDIL Loss__
> As mentioned in Lines 76–84, minimizing CE loss involves continuously increasing the probability of the correct label over the other labels, making the classifier overconfident. Such overconfidence makes it an unreliable OOD detector. Therefore, this motivated us to design a loss function that (1.) encourages desired ranking, and (2.) becomes zero once desired ranking is achieved, to avoid overconfidence.
>
> Moreover, we empirically analyze CE-loss and IDIL loss in section 5.1. We plot FPR vs training steps in figure 3b for CE-loss and IDIL loss and we observe IDIL loss minimization is more correlated with OOD detection performance than CE loss minimization. We also plot their training trajectories in figure 3c and observe that they diverge since the beginning. We leave the theoretical comparison between CE-loss and IDIL loss for future work.
>
> __Qualitative Analysis__
> We present a few examples where SELFOOD accurately detected OOD samples and CE-loss failed. We consider Yelp as in-distribution and TREC as the OOD dataset with BERT classifier. Yelp is a sentiment classification dataset with reviews as text and TREC is a question classification dataset.
>
> A sample from Yelp dataset where CE-loss model gave low scores and SELFOOD gave high scores:
>
> *Its a tequila house - wouldn't you want your staff to be knowledgeable about tequila? Call me silly So Gregg and I are wondering Bourbon Street, looking for the next "place". I don't normally do tequila by itself (bad memories), but I thought, why not. We walk in, and i ask the bartender what the best tequila was. I got the deer in the headlights look. I asked what the most expensive tequila was. She told me it was 80 and pointed to a top shelf with a really cook looking bottle, but she couldn't tell me the name. I then asked what a 35-40 tequila was. She got down a bottle (which took awhile), Gregg and I tried it, I can't say that it was so good and memorable that I can tell you the name. You might want to stick to the frozen margarita machines at this place. I think the bottles are just for show.*
>
> The sentiment of the above review is difficult to predict and the CE-loss model returns low maximum softmax probability scores. However, the SELFOOD model accurately detects its content and generates high scores.
>
> Few samples from TREC dataset where CE-loss gave high scores and SELFOOD gave low scores:
> 1. *Who said "What contemptible scoundrel stole the cork from my lunch ?"*
>
> 2. *Who were the "filthiest people alive ?"*
>
> These samples have strong sentiment-indicative words such as “contemptible scoundrel” and “filthiest people” and CE-loss model predicts high scores. However, SELFOOD recognizes it as not relevant to Yelp and predicts OOD by generating low scores. We will add this discussion to the paper.
>
> __SELFOOD for continuous labels__
> The proposed SELFOOD is not applicable for continuous labels and we leave this for future work. We agree that it would make it more generalised.

---

### Official Review · Reviewer_NgSb · 2023-08-14

**Soundness:** 4

**Excitement:**

3: Ambivalent: It has merits (e.g., it reports state-of-the-art results, the idea is nice), but there are key weaknesses (e.g., it describes incremental work), and it can significantly benefit from another round of revision. However, I won't object to accepting it if my co-reviewers champion it.

**Paper Topic And Main Contributions:**

This work proposes to use the learning to rank method for out-of-distribution detection with only in-distribution data. Specifically, this work designed an IDIL training loss (as well as modified backward propagation) based on pairwise ranking loss. Experiments conducted on well-known datasets such as Yelp and TREC show that this method obtains significant effectiveness improvement compared to existing methods.

**Questions For The Authors:**




**Reasons To Accept:**

This work is well-motivated, and the proposed IDIL training process is clearly defined. It seems simple and effective, providing a promising solution for OOD detection.
The ablation study is comprehensive, including different scenarios. The limitations of the method are also clearly discussed and analyzed.
The paper is well-written and easy to follow. The author also claims to release the code.

**Reasons To Reject:**

Previous work that having similar self-supervised setting should be discussed with more details.
e.g. “Zhou et al.; Zeng et al. introduce self-supervised approaches to OOD detection using contrastive learning”.
From the ablation study, I can see the effectiveness affected by how to perform backward propagation a lot. How do previous work performs if their method also involving backward propagation control? For example, what if rather than pairwise learning to rank loss, the model is trained by listwise infoNCE loss but drop the backward propagation of the positive example term? It is worth to know how important the “pairwise” setting is.


**Reproducibility:**

4: Could mostly reproduce the results, but there may be some variation because of sample variance or minor variations in their interpretation of the protocol or method.

**Reviewer Confidence:**

3: Pretty sure, but there's a chance I missed something. Although I have a good feel for this area in general, I did not carefully check the paper's details, e.g., the math, experimental design, or novelty.

**Typos Grammar Style And Presentation Improvements:**

line 137 ood→OOD

Is the OOD detector usually expected to be an effective classifier at the same time? maybe a little bit more discussion on this.
It would help reader better understand limitation & future direction.

---

> ### Author Rebuttal · Authors · 2023-08-29
>
> We thank the reviewer for the review. Below, we address the reviewer’s main concerns:
>
> __More discussion on self-supervised approaches__
> Zhou et al propose an unsupervised training method to fine-tune transformer models on a combination of contrastive loss and cross entropy loss. They experiment with several scoring functions such as maximum softmax probability, energy score, mahalanobis distance, and cosine similarity. Zeng et al. also propose an unsupervised training approach using supervised contrastive Loss and cross Entropy loss to train Bi-LSTM models for OOD Detection. They employ the BERT model only to obtain encodings that are fed into the BiLSTM model. SELFOOD follows a similar experimental setting as both, however during training, we do not use cross entropy loss and backpropagate gradients to decrease probabilities associated with incorrect labels rather than increasing the probability of the correct label, which results in faster convergence. Moreover, our method is also applicable to transformer-based architectures. We will add more discussion and comparison with these two papers.
>
> __Effect of Back propagation control on other losses__
> In order to understand the importance of back propagation control on other losses, we experiment with Triplet Margin Loss. We train a BERT-based OOD detector with TREC as in-distribution datasets and evaluate on Clima, NYT, Yelp as the OOD datasets. The embedding vectors from the trained detector are used to identify the OOD samples using Mahalanobis distance. We run two settings i.e. with and without dropping the backpropagation through the positive term, to compare the improvement. The results are shown below.
>
> | **OOD Dataset**  | **Method**                         | **FPR95**    | **AUROC** |
> |------------------|------------------------------------|--------------|-----------|
> | Clima          | Triplet Marging Loss Grad Neg Only | 0.0698       | 97.916    |
> | Clima          | Triplet Marging Loss Grad Both     | 6.2067       | 99.219    |
> |                  |                                    |              |           |
> | NYT              | Triplet Marging Loss Grad Neg Only | 0.33208      | 99.1708   |
> | NYT              | Triplet Marging Loss Grad Both     | 1.7030e-2    | 99.9941   |
>
>
> We do not notice any significant improvement by dropping the backpropagation through the positive term. We hypothesize that contrastive losses such as Triplet Margin Loss work on bringing together intra-label data points in the latent space while the IDIL loss aims to modify the softmax probabilities of the data points with an additional linear head.
>
> __OOD detector as a separate classifier__
> Several works previously trained a separate OOD classifier which is different from the classifier for the underlying task. For example, [1] propose a Random Forest OOD detector as an external calibrator in addition to a Question Answering model. [2] experiment with the mahalanobis distance for OOD detection and compare its effect across multiple models. They conclude that although mahalanobis distance alongside transformer based models is effective, such methods are sensitive to the geometric features in the embedding space and can be spoilt if the embedder is used for classification and is overfit. This matches with our observation in Lines 523-526, where upon adding intra-document comparison to the loss, the OOD detection performance drops. Therefore, we believe training an OOD classifier separately is a good design. Balancing the inter-document and intra-document comparisons is crucial for achieving optimal performance in both OOD detection and text classification tasks and we leave this for future work.
>
> [1] Selective Question Answering under Domain Shift ACL’20
>
> [2] Revisiting Mahalanobis Distance for Transformer-Based Out-of-Domain Detection AAAI’21

---

### Meta-Review · Area_Chair_3sWo · 2023-09-19

**Recommendation:** 2

**Metareview:**

This work proposes to use the learning-to-rank method for out-of-distribution detection with only in-distribution data. The method being proposed is mainly used to lower the desired confidence ranking by decreasing the probabilities associated with incorrect labels. The paper is well-written and well-motivated. However, as all the reviewers pointed out, the paper is restricted to a single task of text classification without the generalization to any other NLP tasks. This limitation decreases the impact of this paper.

---

### Decision · Program_Chairs · 2023-10-07

**Decision:**

Accept-Findings

**Comment:**

This work proposes to use the learning-to-rank method for out-of-distribution detection with only in-distribution data. The method being proposed is mainly used to lower the desired confidence ranking by decreasing the probabilities associated with incorrect labels. The paper is well-written and well-motivated. However, as all the reviewers pointed out, the paper is restricted to a single task of text classification without the generalization to any other NLP tasks. This limitation decreases the impact of this paper.